Hawaiian black coral (Antipatharia) complete mitochondrial genomes have limited phylogenetic signal for taxonomic resolution of species

Wishingrad Van 1
http://orcid.org/0009-0000-8280-7198 Shizuru Leah E. K. 1
Takata Kenji 1 2
http://orcid.org/0000-0002-3430-3451 Montgomery Anthony D. 3
Wagner Daniel 4
http://orcid.org/0000-0001-6339-4340 Toonen Robert J. 1 rjtoonen@gmail.com
1 Hawai‘i Institute of Marine Biology, University of Hawaii at Manoa , Kaneohe, Hawai‘i , United States
2 Graduate School of Agricultural and Life Sciences, University of Tokyo , Tokyo, Yayoi , Japan
3 Pacific Fish and Wildlife Office, US Fish and Wildlife Service , Honolulu, Hawai‘i , United States
4 Ocean Exploration Trust , Honolulu, Hawai‘i , United States
Wang Liang
Electronic publication date: 2025 May 30
Publication date: 2025
Volume: 13
Electronic Location ID: e18731
Received 2024 Aug 7; Accepted 2024 Nov 27
Copyright: © 2025 Wishingrad et al.
Copyright year: 2025
Copyright holder: Wishingrad et al.
License: This is an open access article distributed under the terms of the Creative Commons Attribution License, which permits unrestricted use, distribution, reproduction and adaptation in any medium and for any purpose provided that it is properly attributed. For attribution, the original author(s), title, publication source (PeerJ) and either DOI or URL of the article must be cited.
License URL: https://creativecommons.org/licenses/by/4.0/

Keywords: Antipathes, Aphanipathes, Cirrhipathes, Hexacoralia, Mesophotic zone, Myriopathes, Phylogenetics, Stichopathes, Shallow-water

Funding: Joint Institute for Marine and Atmospheric Research NA16NMF4320058, NA17NMF4320250, NA17NMF4320293, and NA17NMF4320294 National Science Foundation OA#1416889 and NSF OCE#2048457 U.S. Fish and Wildlife Service Council for Native Hawaiian Advancement, the HIMB Hayashida Endowment, and the Schmidt Scholarship This research was supported by the NOAA Office of Habitat Conservation, Pacific Islands Fisheries Science Center Deep-sea coral habitat program, facilitated through the Joint Institute for Marine and Atmospheric Research under cooperative agreements NA16NMF4320058, NA17NMF4320250, NA17NMF4320293, and NA17NMF4320294, as well as funding from the National Science Foundation under grant numbers OA#1416889 and NSF OCE#2048457. The findings and conclusions in this article are those of the author(s) and do not necessarily represent the views of the U.S. Fish and Wildlife Service. Financial support from the Council for Native Hawaiian Advancement, the HIMB Hayashida Endowment, and the Schmidt Scholarship played a vital role in the success of this work. The funders had no role in study design, data collection and analysis, decision to publish, or preparation of the manuscript.

==============================
Most inferences about black coral (Antipatharia) phylogenetics have relied on a handful of molecular markers from PCR-Sanger methods but recently complete mitogenomes are shedding additional light on relationships. We present the most complete survey of shallow-water to mesophotic Hawaiian black corals (‘ēkaha kū moana) to date based on complete mitogenome sequences. The phylogenetic relationships inferred from whole-mitochondrial phylogenies recover Myriopathidae as monophyletic with Myriopathes and Tanacetipathes as the outgroups to all other Hawaiian black coral taxa. Combining our data with other published mitochondrial datasets for black corals, we find that morphologically similar Cirrhipathes cf. anguina specimens are divergent and may not be conspecifics. Likewise, the genera Antipathes and Stichopathes (family Antipathidae) include species that are more divergent from one another than they are to other genera in family Aphanipathidae. Overall, data show Myriopathidae is a monophyletic family, but the families Aphanipathidae and Antipathidae are polyphyletic, and the genera Antipathes and Stichopathes live up to their reputation as a “taxonomic dumping ground”. These phylogenetic analyses underscore the need for continued research to understand the evolutionary history and phylogenetic relationships for black corals generally and ‘ēkaha kū moana specifically.

Introduction

Antipatharians, commonly known as black corals, are a globally distributed, slow-growing, and long-lived order of hexacoral cnidarians (Daly et al., 2007; Bo, 2008; Wagner, Luck & Toonen, 2012; Brugler, Opresko & France, 2013; Shizuru, 2023)1 . These colonial suspension feeders are characterized by small, non-retractile polyps with six unbranched tentacles and a spiny, proteinaceous skeleton (Opresko, 1972; France, Brugler & Opresko, 2007; Molodtsova & Budaeva, 2007; Wagner, Luck & Toonen, 2012; Wagner, 2015). They occupy a broad bathymetric range (Bo et al., 2012; Wagner, Luck & Toonen, 2012; Hitt et al., 2020), from shallow-water at 3.4 m (Agarwal et al., 2024) to the deepest records observed at 8,600 m (Molodtsova, Sanamyan & Keller, 2008; Yesson et al., 2017). Antipatharians exhibit diverse growth forms, from unbranched to branched into a bush, fan, feather, or bottle-brush morphology. Their tissues display various colors, including brown, red, orange, pink, yellow, green, white, and gray; all these characters have been proposed to be taxonomically informative in some taxa (Wagner, Luck & Toonen, 2012).

Antipatharians are ecologically important because they form an important structural framework of living benthic habitats between 50–200 m and represent a foundational species group that increases habitat complexity at these depths (Wagner, Luck & Toonen, 2012). About 63% of antipatharian genera documented to date globally are found at mesophotic depths between 30–150 m (Bo et al., 2019). As ecosystem engineers, they significantly enhance biodiversity by creating a habitat within which various other organisms strongly associate (Criales, 1980; Boland & Parrish, 2005; Love et al., 2007; Molodtsova & Budaeva, 2007; Wagner, Luck & Toonen, 2012; Gress & Kaimuddin, 2021; Brickner et al., 2022; Ávila-García et al., 2023; Gonzalez, Conde-Vela & Osborn, 2023). In Hawai‘i, black corals (‘ēkaha kū moana, which translates to “fern of the sea”) are also of great cultural importance. The Kumulipo, the Hawaiian creation story, begins with the emergence of the coral (ko‘a) from darkness and continues through the creation of the Hawaiian Islands, plants, animals, and humans. It is widely believed that ‘ēkaha kū moana symbolizes the ko‘a mentioned in the opening lines of the Kumulipo because ‘ēkaha kū moana are typically found in lightless ocean depths. Kānaka ‘ōiwi (native Hawaiians) thus hold great reverence for the ko‘a as their ancestral progenitor.

In addition to their ecological and cultural importance, black corals are prized for their medicinal products and inherent natural beauty. Numerous cultures across the globe use black corals for medicinal purposes, as well as to produce jewelry and art. The name Antipatharia comes from the Greek words (“anti” and “pathos”) that literally translate to “against disease”, and ancient Hawaiians used ‘ēkaha kū moana for medicinal purposes (Kaaiakamanu & Akina, 1922; Wagner, Luck & Toonen, 2012). Furthermore, ‘ēkaha kū moana are the official gemstone of the State of Hawai‘i and support a multi-million-dollar fishery employing over 500 individuals in manufacturing and retail statewide (Grigg, 2001). Established in 1958, the fishery was long touted as sustainable with population stability of Antipathes griggi (formally referred to as A. dichotoma) due to effective management and a depth refuge below which harvest did not occur (Grigg, 2001). Subsequent taxonomic revision revealed this fishery harvested three species: Antipathes griggi (formerly A. dichotoma), Antipathes grandis, and Myriopathes cf. ulex (Opresko, 2009; Wagner et al., 2017). As primary suppliers, SCUBA divers harvest these corals from depths 40 to 70 m, primarily within the ‘Au‘au Channel between Lāna’i and Maui. Additionally, albeit to a lesser extent, divers collect corals off the coasts of Kaua‘i and Hawai‘i (Wagner et al., 2017). Surveys up until the 1990s indicated that A. griggi, A. grandis, and M. cf. ulex comprised roughly 90%, 10%, and 1% of the harvests respectively (Oishi, 1990). Surveys through 1998 indicated that rates of recruitment and growth remained steady, leading to the conclusion that the populations were stable (Grigg, 2001). However subsequent surveys showed a marked decline in black coral standing stocks compared to earlier observations, often attributed to harvesting and overgrowth by an alien invasive octocoral (Grigg, 2004).

The depth refuge from harvest that was once believed to exist was brought into question with more detailed taxonomic evaluations of ‘ēkaha kū moana. For example, colonies of A. griggi below 90 m in the ‘Au‘au Channel off Maui was described as a new subspecies Aphanipathes verticillata mauiensis (Opresko et al., 2012; Wagner et al., 2017). In fact, it turns out that A. verticillata mauiensis is morphologically similar to A. griggi and the dominant coral in the ‘Au‘au Channel at depths of 88 to 130 m (Opresko et al., 2012; Wagner et al., 2017), which includes most of the harvest depth refuge previously thought to exist for A. griggi. Recent studies on antipatharian depth distributions in the ‘Au‘au Channel found species composition at harvest depths consisted of 93% A. griggi and 7% A. grandis, whereas within the presumed depth refuge in deeper waters from 71–130 m, the composition changed to 68% A. grandis, 25% A. verticillata, and only 7% A. griggi (Wagner et al., 2017). Together, the declines in biomass, revised taxonomic resolution of species and new data on species distributions refute previous claims of a depth refuge resulting in a sustainable fishery and point instead to historical overharvesting. Thus, these studies underscore the susceptibility of A. griggi to prolonged fishing pressure. In addition to the pressures of overfishing, the ecology of ‘ēkaha kū moana, particularly their slow growth rates, makes them particularly susceptible to natural and anthropogenic perturbances (Koslow et al., 2001; Sampaio et al., 2012; Wagner, Luck & Toonen, 2012), and are therefore of great conservation concern.

Despite Hawaiian black corals being among the most studied of antipatharian faunas (Grigg, 1976; Wagner, Luck & Toonen, 2012), identification remains challenging due to the lack of high-quality type specimens, poor species descriptions, and few distinct morphological characters among taxa (Wagner et al., 2010; Wagner, 2015). For example, type materials are missing for Cirrhipathes anguina and M. ulex (Wagner, 2015), and the species descriptions are vague (Ellis & Solander, 1786; Dana, 1846). Recent species documentation of black corals have relied predominantly on taxonomic characters such as colony branching pattern, polyp structure, and skeletal spine morphology, as well as in situ photographs and scanning electron microscopy of skeletal features (Opresko, 2009; Wagner & Opresko, 2015; Opresko & Wagner, 2020). However, plasticity in skeletal traits, both within and across species, dramatically complicates the taxonomy of this order (Bo et al., 2012; Wagner, Luck & Toonen, 2012; Opresko, Nuttall & Hickerson, 2016). Likewise, low genetic variation may provide little to no resolution, even among groups that are highly distinct morphologically (Bledsoe-Becerra et al., 2022). Using a suite of morphometrics, in situ observations, and genetic characters, the taxonomy of ‘ēkaha kū moana has recently been revised (Opresko, 2009; Wagner et al., 2010; Opresko et al., 2012; Wagner, 2015; Wagner & Opresko, 2015; Opresko & Wagner, 2020; Molodtsova, Opresko & Wagner, 2022). Nonetheless, uncertainties about taxonomic affinity and relationships among black coral taxa persist (Wagner et al., 2010; Bo et al., 2012; Barrett et al., 2020; Asorey et al., 2021; Tapia-Guerra et al., 2021), and relatively few studies have used a phylogenomic approach to test species boundaries in these taxonomically challenging corals (e.g., Horowitz et al., 2020, 2023). Here, we present the most complete survey of shallow-water Hawaiian black coral phylogenetics to date with the intent to test species boundaries and resolve relationships among ‘ēkaha kū moana. This will aid in inferring species relationships between these understudied Hawaiian species sequenced here for the first time. Furthermore, we compare these datasets to currently accepted nomenclature to evaluate the status of Hawaiian black coral taxonomy.

Materials and Methods

Sample collection

We selected a group of 10 individuals representing the full range of known morphological phenotypes and type material from species at the Bernice Pauahi Bishop Museum black coral collection (Fig. 1, Table 1). We collected coral fragments by hand via rebreather diving and from the human occupied Hawai‘i Undersea Research Laboratory submersibles Pisces IV and V at depths of 20 to 150 m where Hawaiian black corals are most common (Kahng et al., 2010; Wagner, 2015) following the collection methods described in Wagner (2015). We preserved all tissue samples in 95% ethanol and stored them at room temperature until we performed DNA extraction. These samples include six currently recognized species and encompass the range of observed morphotypes known to occur at these depths in the Hawaiian Archipelago (Wagner, 2015). All specimens were gathered in accordance with the relevant collection permits issued by co-management agencies of the NOAA Papahānaumokuākea Marine National Monument, as well as the State of Hawai‘i Department of Land and Natural Resources, Division of Aquatic Resources, Special Activities Permits #SAP-2008-04 and SAP-2009-13.

Figure 1 Map of the main Hawaiian islands and the Northwest Hawaiian Islands showing the collection sites of the antipatharians used in this study in red.

The following samples were collected from around Maui: Antipathes grandis (#168 and #187), Antipathes griggi (#196), Aphanipathes verticillata (#130), and Myriopathes ulex (#144). Antipathes griggi (#176) and Stichopathes cf. maldivensis (#180) were collected from Kaua’i, Cirrhipathes cf. anguina (#361) was collected at Lalo, and Myriopathes cf. ulex (#269) was collected from Manawai. See Table 1, for more information—map courtesy of NOAA Papahānaumokuākea Marine National Monument.

Table 1 ‘Ēkaha kū moana (Hawaiian Black Coral) samples used in this study.

Antipathes griggi #196 represents type material stored at the Bernice Pauahi Bishop Museum (BPBM).

Species name	Voucher ID	Island	Site	Date	Depth (m)	Genome size (bp)	Comments	
Antipathes grandis	168	Maui	Keyhole Pinnacles, ‘Au‘au	22-Feb-09	100	20,484		
Antipathes grandis	187	Maui	‘Au‘au Channel	07-Apr-09	102	20,484	A. grandis w/red polyps	
Antipathes griggi	176	Kauai	Amber‘s Arches, S. Kauai	04-Mar-09	23	20,460	A. griggi w/thick polyps	
Antipathes griggi	196	Maui	‘Au‘au Channel	07-Apr-09	93	20,462	Type: BPBM-D1879	
Aphanipathes verticillata	130	Maui	‘Au‘au Channel	04-Apr-08	111	20,395		
Cirrhipathes cf. anguina LS-2022	179	Kauai	Amber‘s Arches, S. Kauai	04-Mar-09	23	20,452	From Shizuru et al. (2024); yellow polyps	
Cirrhipathes cf. anguina	361	French Frigate Schoals	French Frigate Schoals	16-Aug-10	30	20,462		
Myriopathes ulex	144	Maui	‘Au‘au Channel	05-Apr-08	96	17,711		
Myriopathes cf. ulex	269	Pearl & Hermes	off Pearl & Hermes Atoll	17-Aug-09	61	17,711		
Stichopathes cf. maldivensis	180	Kauai	Amber‘s Arches, S. Kauai	04-Mar-09	23	20,460		

DNA extraction and quantification

Genomic DNA isolations were accomplished with a modified protocol for the Omega Bio-Tek (Norcross, GA, USA) E-Z 96 Tissue DNA Kit as follows. We first conducted a rapid elution with 100 µl with high-performance liquid chromatography (HPLC)-grade water to selectively eliminate small fragments of degraded DNA. We then performed a second elution using 100 µl of HPLC-grade water, incubated for 5 min at 70 °C, to recover the remaining high molecular weight DNA. We evaluated extraction quality by visually examining genomic extractions on a 2% agarose gel. We stained genomic DNA with GelRed (Biotium, Fremont, CA, USA) and evaluated size relative to a 1 kb DNA Hyperladder I (New England Biolabs, Ipswich, MA, USA). We proceeded with samples that had a high molecular weight band or a smear with at least half of the total genomic DNA above 2,500 bp (following Knapp et al., 2016; Johnston et al., 2017). We quantified DNA concentrations using the AccuBlue High Sensitivity dsDNA (Biotium, Fremont, CA, USA) or Qubit dsDNA High Sensitivity (Invitrogen, Waltham, MA, USA) quantification kits.

Library preparation and sequencing

We generated ezRAD reduced-representation genomic libraries (Toonen et al., 2013) following the protocol of Knapp et al. (2016). Briefly, we digested genomic DNA with the isoschizomers MboI and Sau3AI (New England Biolabs, Ipswich, MA, USA), which recognize 5′ GATC cut sites. We digested a total of 375 ng of genomic DNA from each sample using 2 μl of each MboI and Sau3AI in a 50 μl reaction volume following manufacturer protocols. We incubated each digestion in a thermocycler at 37 °C for 3 h, followed by inactivation at 65 °C for 20 min. Digestion was verified visually by running each digest beside undigested genomic DNA on a 1% agarose gel. DNA fragments were size-selected (300–600 bp) using a Pippen Prep (Sage Science, Beverly, MA, USA) before preparing genomic libraries for sequencing using either the Illumina TruSeq Nano (Illumina, San Diego, CA, USA) or Watchmaker DNA Library Preparation (Watchmaker Genomics, Boulder, CO, USA) kits. Following bioanalyzer and qPCR control checks, libraries were sequenced on the Illumina MiSeq (V3 2 × 300 bp paired-end) at the IIGB Genomics Core facility at UC Riverside or the Advanced Studies in Genomics, Proteomics and Bioinformatics (ASGPB) at the University of Hawai‘i at Mānoa.

Mitochondrial genome assembly

Mitochondrial genome assembly followed previously published protocols (Forsman et al., 2017; Shizuru et al., 2024). Briefly, we used TRIM GALORE! v. 0.6.0 (Krueger, 2015) to filter and trim low-quality reads and remove Illumina adapters. We again trimmed any ends in which Q-scores dropped below 20 before removing the first 13 bp of the standard Illumina paired-end adapters (‘AGATCGGAAGAGC’). We performed de novo mitogenome assembly using SPAdes v. 3.13.0 (Bankevich et al., 2012). For each de novo assembly, we first created a database containing all contigs >10,000 bp, then used BLAST to confirm that only antipatharian sequences were included (see code Supplemental Data). We circularized contigs with the highest percent identity and query cover to antipatharians and trimmed overlapping ends in Geneious Prime 2022.1.1 (https://www.geneious.com). We used published antipatharian mitogenomes from ‘Cirrhipathes’ (Stichopathes, see below) luetkeni (Kayal et al., 2013); Stichopathes sp. SCBUCN-8849 (Asorey et al., 2021), Stichopathes sp. SCBUCN-8850 (Asorey et al., 2021), Trissopathes cf. tetracrada NB-2020 (Barrett et al., 2020), and Chrysopathes formosa (Brugler & France, 2007) to identify protein-coding regions using the live annotate feature in Geneious Prime 2022.1.1. We note here that the taxonomic name Cirrhipathes luetkeni (JX023266, Kayal et al., 2013) is invalid as a superseded combination, and the proper identification for this species should be Stichopathes luetkeni (WoRMS, 2023).

Phylogenetic analyses

We aligned annotated regions of the mitogenome using the MAFT v7.490 plugin in Geneious Prime with the L-INS-I algorithm, using a scoring matrix of 200PAM/k = 2, a gap open penalty of 1.53, and an offset value of 0.123. We concatenated gene alignments of the mitogenomes that we generated and included these along with other published antipatharian mitogenomes to determine phylogenetic relationships with closely related taxa (sensu Barrett et al., 2020). Additionally, we generated phylogenetic trees for each mitochondrial gene separately to compare individual gene trees to the concatenated dataset. For our concatenated, partitioned mitogenome dataset, we used IQTREE v.2.0.3 (Minh et al., 2020; Chernomor, Von Haeseler & Minh, 2016). We used ModelFinder (Kalyaanamoorthy et al., 2017) to determine the best-fit substitution model for individual gene trees and the concatenated dataset.

Results

Mitochondrial genomes

There is considerable disparity in mitogenome size, ranging from 17,711 to 20,484 bp due primarily to variation in the length of intergenic regions among taxa (Figs. S8–S16). Like other hexacorallia anthozoans, these mitogenomes contain 13 protein-coding genes (ATP6, ATP8, COX1-3, CYTB, ND1-6), two rRNA genes (rnl and rns), and two tRNA genes (trnM and trnW) (Brugler & France, 2007; Sinniger, Chevaldonné & Pawlowski, 2007; Barrett et al., 2020; Asorey et al., 2021; Tapia-Guerra et al., 2021). All taxa include a cytochrome c oxidase subunit I (COX1) intron and a group 1 intron in NADH dehydrogenase subunit 5 (ND5). All but Aphanipathes verticillata (#130), Myriopathes ulex (#144), and Myriopathes cf. ulex (#269) contain an embedded homing endonuclease gene (HEG, Tables S2–S8; Barrett et al., 2020). Complete mitochondrial sequences are available on NCBI (see DNA Deposition section and Table S1 for list of accession numbers) and raw data are available on NCBI under BioProject accession PRJNA1177630.

Mitochondrial phylogenies

ModelFinder identified TVM+F+G4 as the best-fit model for our concatenated mitogenome dataset. A maximum likelihood phylogeny of 38 Antipatharian species from seven families shows that all Myriopathes species form a major clade with Tanacetipathes to support a monophyletic grouping of the family Myriopathidae (Fig. 2, clade A). The second major clade recovered (clade B) splits into two subclades. The first (clade B1) contains Phanophathes sp. NB-2020, Stichopathes sp. NB-2020, Stichopathes abyssicola, and Antipathes cf. dichotoma NB-2020 whereas the other (clade B2) contains a mixture of Cirrhipathes, Stichopathes, and Antipathes species as closely related with a relatively long branch to the lone Aphanipathes verticillata sample (Fig. 2). We rooted the tree using a zoanthid (Zoanthus sansibaricus) and an anemone (Metridium senile), and recover Leiopathes as the black coral outgroup to all other black corals. All single-gene phylogenies were similar, though not identical (Figs. S1–S7). Among all 13 loci, ND5 was the most variable gene region, with 77.1% pairwise identity and 36.1% identical sites. This variability suggests that this region could be most informative for future population genetic analyses that rely on mitochondrial data. Although not all gene trees are concordant, maximum likelihood phylogenetic reconstruction based on both ND4 and COX1 (Figs. S4 and S6) are consistent with the mitogenome phylogeny based on all 13 protein-coding mitochondrial genes when aligned and concatenated (Fig. 2). This phylogenetic tree places Aphanipathidae and Antipathidae as closely related, with Myriopathidae as sister to both groups (Fig. 2). Overall, our results are incongruent with currently accepted taxonomy because the families Antipathidae and Aphanipathidae are polyphyletic, as are the genera Antipathes, Cirrhipathes, and Stichopathes. For example, both Antipathes and Stichopathes include species that are more divergent from one another than either is to a species in the family Aphanipathidae. Further, there is strong support for Clades B1 and B2 (SH-aLRT= 100 ultrafast bootstrap = 100) (Fig. 2). Still, these clades each contain members of both Aphanipathidae and Antipathidae, leaving Myriopathidae as the only family that currently appears monophyletic in our dataset of the Hawaiian ‘ēkaha kū moana. However, Tylopathes sp. from the family Stylopathidae closely groups with Myriopathidae as well, though it is unclear if Tylopathes is within, or sister to, the Myriopathidae clade (Fig. 2).

Figure 2 Phylogenetic tree summarizing relationships among Hawaiian black coral taxa.

Highlighted taxa show family-level relationships as they are currently defined. This tree represents a maximum likelihood phylogeny of 38 antipatharian taxa and two outgroups estimated in IQ-TREE and inferred from 13 individually aligned and concatenated protein-coding mitochondrial genes (see text for details). The mitogenomes sequenced from Hawai’i are bolded (See Table 1 for collection sites). Antipathes griggi (#196) is the holotype stored at the Bernice P. Bishop Museum. Branch lengths are relative to genetic divergence, where longer branches indicate more diverged, and values at each node represent SH-aLRT/ultrafast bootstrap values, where higher numbers indicate higher confidence.

Discussion

Despite the ecological, cultural, and economic importance of antipatharians across the globe, their phylogenetic relationships and taxonomy remain understudied and poorly understood. Here, we provide the first complete phylogenomic examination of relationships among shallow-water ‘ēkaha kū moana (Hawaiian black coral species) using mitochondrial genomes of all available species and ecotypes in this region. Our analysis of all the known diversity of Hawaiian black corals from shallow-water to mesophotic zones is consistent with previous studies based on mitogenomes (Barrett et al., 2020) and ultra conserved elements (UCEs; Horowitz et al., 2020, 2023) in finding that currently recognized nominal groups are not corroborated by either mitochondrial or nuclear phylogenetic data and are clearly in need of taxonomic revision.

Morphological identification (following Wagner, 2015) places these Antipatharians species into three families—Antipathidae, Aphanipathidae, and Myriopathidae, five genera—Antipathes, Aphanipathes, Cirrhipathes, Myriopathes, and Stichopathes—and six species—Antipathes grandis, Antipathes griggi, Aphanipathes verticillata, Myriopathes ulex, Myriopathes cf. ulex, and Stichopathes cf. maldivensis (Bo et al., 2012 (clade D); Terrana et al., 2020; Bacharo & Sotto, 2022). Diagnostic traits used to define taxonomic groups include differences in branching pattern, polyp morphology, thickness of the stem, or color (Wagner, 2015). As defined on the basis of morphology, corals belonging to the family Aphanipathidae possess obscure polyps (0.5–1.3 mm in transverse diameter), hence its Greek root “aphano” which translates to “invisible” (Opresko, 2004; Wagner, 2015). Further, this family is divided into two subfamilies based on polypar spine development: Aphanipathinae and Acanopathinae (Opresko, 2004). Corals that belong to Aphanipathinae have skeletal spines of similar heights on the side of the corallum where the polyps exist (Opresko, 2004; Wagner, 2015). In contrast, Acanthopathinae presents spines directly below the oral opening and on the outer edges of polyps (Opresko, 2004; Wagner, 2015). Myriopathidae, which comes from the Greek word “myriophylla,” signifying many branches, is named for the extensive branching displayed by the corals belonging to this family (Opresko, 2001; Wagner, 2015). Corals in this family contain polyps with six primary and four secondary mesenteries that are 0.5–1.0 mm in transverse diameter. Corals in the Myriopathidae family have short, rounded tentacles (Opresko, 2001; Wagner, 2015). Morphologically, the family Antipathidae does not have well-defined characters that distinguish them from other families, but generally have polyps that (1) are approximately 0.5 to 1.0 mm in transverse diameter (with the notable exception of whip or wire coral in the genus Cirrhipathes which has polyps reported to reach 3.9 mm in diameter (Terrana et al., 2020)), (2) are short in the transverse plane, (3) consist of 10 mesenteries (six primary and four secondary), and (4) have two tentacles perpendicular to the branch bearing polyp that are longer than the four tentacles adjacent to the branch bearing polyp (Opresko & Sanchez, 2005; Bo, 2008; Moon & Song, 2008; Wagner, 2015). Although specific genera exhibit clear morphological distinctions, such as the unbranched Cirrhipathes (polyps arranged irregularly on all sides of the corallum), unbranched Stichopathes (polyps arranged in a single row on one side of the corallum), Allopathes which have multiple elongated stems arising from the base (Opresko & Cairns, 1994), and Blastopathes which exhibits stem-like branches (Horowitz et al., 2020), most of the remaining genera are challenging to differentiate (but also note Allopathes and Blastopathes have not been reported in Hawai‘i). Antipathidae (Ehrenberg, 1834) encompasses such diverse genera as Antipathes (Pallas, 1766), Cirrhipathes (Blainville, 1857), and Stichopathes (Brook, 1889). In particular, the genus Antipathes has historically been considered a “taxonomic dumping ground” due to the variety and kinds of characters used to distinguish species in this genus (Daly et al., 2007; Brugler, Opresko & France, 2013; Wagner, 2015; Bo et al., 2018). Though some species have been reclassified based on colony branching pattern and removed from Antipathes (such as those with systematically arranged pinnules now classified as Myriopathes, Opresko, 2001; or deep-water (>1,000 m) Stichopathes now reclassified as Aphanostichopathes, Opresko & Molodtsova, 2021), uncertainties remain about taxonomic affinities within this genus (Daly et al., 2007; Bo, 2008; Wagner, 2015; Tapia-Guerra et al., 2021). Likewise, several genera have been re-classified and removed from Antipathidae, and uncertainty remains about genera still grouped into this family (Daly et al., 2007; Bo, 2008; Wagner, 2015; Tapia-Guerra et al., 2021). Visual identification of these groups is ultimately proving to not be reliable.

Previous authors have found mitogenomic data do not support existing morphologically-based taxonomy, and others have highlighted the need for taxonomic revision of antipatharians (Daly et al., 2007; Wagner et al., 2010; Bo et al., 2012; Wagner, 2015; Bo et al., 2018; Barrett et al., 2020; Asorey et al., 2021; Tapia-Guerra et al., 2021). Here, we confirm with molecular data that Myriopathes cf. ulex is closely related to Myriopathes ulex and that the family Myriopathidae appears to be monophyletic based on our sampling to date. In contrast, the families Aphanipathidae and Antipathidae are polyphyletic, and the genera Antipathes, Cirrhipathes, and Stichopathes live up to their reputation as a “taxonomic dumping ground” (Daly et al., 2007; Brugler, Opresko & France, 2013; Wagner, 2015; Bo et al., 2018). Both the genera Antipathes and Stichopathes (family Antipathidae) include species that are more divergent from one another than either is to species in the family Aphanipathidae. Furthermore, the family Antipathidae has been described as a taxonomic catch-all following recent phylogenetic reconstructions comparing available morphological and genetic data (Daly et al., 2007; Bo, 2008; Wagner, 2015). Thus, our study reinforces the need for taxonomic revision at all levels within the Hawaiian antipatharians.

Mitogenome size has been proposed as a taxonomic character for some taxa (e.g., Kong et al., 2020; Duminil & Besnard, 2021), and widespread differences among Hexacorallia in mitogenome size due to intergenic regions are known from Antipatharia, Zoantharia, Actinaria, and Scleractinia (Barrett et al., 2020; Feng et al., 2023; Lin et al., 2014; Medina et al., 2006; Sinniger, Chevaldonné & Pawlowski, 2007). We find considerable disparity in mitogenome size among taxa, but it remains unclear to what extent mitogenome size matches accepted taxonomy. The first mitogenome sequenced from a Hawaiian black coral was Cirrhipathes cf. anguina LS-2022 (Shizuru, 2023) which is highly similar (99.97%) to that previously reported for Stichopathes sp. (MZ157400) from Rapa Nui (Asorey et al., 2021). These two samples were collected nearly 8,000 km apart (Rapa Nui and Hawai‘i) and identified morphologically as belonging to different genera, but it seems unlikely that intergeneric taxa share such similar mitogenomes. ITS1-based reconstructions place Stichopathes sp. SCBUCN-8849 within a clade primarily within the genus Cirrhipathes (Asorey et al., 2021). Interspecific mitogenomes sequenced to date among Stichopathes differ by 0.9–2.2%, which is roughly the same magnitude of difference by which these species differ from the mitogenomes of Antipathes (Fig. 2; Asorey et al., 2021). Here we add nine additional antipatharian mitogenomes to the 29 published previously, including the first representative from the genus Aphanipathes. Based on our limited sample size, it appears that Myriopathidae tend to have a smaller mitogenome (~17 kbp) than the others (~20 kbp). Still, with a low sample size and only a single representative of Aphanipathidae, it is not possible to say whether mitogenome size may be a diagnostic character, but this remains an intriguing possibility.

One notable characteristic of antipatharians and zoantharians that distinguishes them from other Hexacorallia is the general absence of gene rearrangement (Barrett et al., 2020). Regardless of variation in size due to intragenic regions, the mitogenomes of ‘ēkaha kū moana displayed a similar arrangement of genes across families and showed conservation of gene order and content relative to other antipatharians (Brugler & France, 2007; Kayal et al., 2013; Barrett et al., 2020; Asorey et al., 2021; Tapia-Guerra et al., 2021). Like many other Hexacorallia, two introns are present in all ‘ēkaha kū moana species: one within the cytochrome c oxidase subunit I (COX1) and another within the NADH dehydrogenase subunit 5 (ND5) (Sinniger, Chevaldonné & Pawlowski, 2007; Barrett et al., 2020). Interestingly, while we find ND5 is the most variable gene region among these species, (Barrett et al., 2020) identified ND4 as the most variable gene region among other antipatharian species. Moreover, Horowitz et al. (2020) found the nad5-IGR-nad1(igrN) region of the mitochondrial genome recovered a similar topology to that of UCEs, suggesting ND4 and ND5 are phylogenetically informative regions. Like many orders of Cnidaria (Actinaria, Corallimorpharia, Scleractinia, Zoanthidea, Gorgonacea, Alyconacea, and Hydroida) and the antipatharian families Schizopathidae and Cladiopathidae (Goddard et al., 2006; Barrett et al., 2020), Aphanipathes verticillata (#130), Myriopathes ulex (#144), and Myriopathes cf. ulex (#269) lack an embedded homing endonuclease gene (HEG). In contrast, Phanopathes sp. NB-2020, Stichopathes sp. n. NB-2020, and Antipathes cf. dichotoma NB-2020 all possess an embedded HEG despite their intermediate placement on the tree relative to these three Hawaiian black coral samples. Given such high gene conservation across taxa, the shared presence or absence of an embedded HEG among these ‘ēkaha kū moana (currently in different families) seems noteworthy and may help shed light on evolutionary relationships. Given polyphyly at the family level and limited resolution at the species level for ‘ēkaha kū moana based on mitogenome sequences, it will likely require genomic-wide sequencing to resolve these relationships.

Conclusions

The combined ecological, economic, and cultural value of ‘ēkaha kū moana highlights the importance of essential taxonomic work to resolve their phylogenetic relationships, understand species distributions within the Hawaiian Archipelago, and shed light on their population dynamics. Moreover, differentiating between species is critical for understanding population trends for taxa targeted in this fishery. However, we find limited phylogenetic signal for taxonomic resolution, where Myriopathidae is a monophyletic family, but the families Aphanipathidae and Antipathidae are polyphyletic, and the genera Antipathes and Stichopathes live up to their reputation as a “taxonomic dumping ground”. The recent discovery that there is no depth refuge for the most targeted species, as once believed, raised significant concerns about the long-term sustainability of the ‘ēkaha kū moana fishery (Opresko et al., 2012; Wagner et al., 2017). In addition to the ecological and economic impacts, overfishing of ‘ēkaha kū moana carries profound cultural importance to the Hawaiian people as evidenced by their descriptions in the Kumulipo, adding to the conservation value of this group. Continued contributions to our understanding of black coral genomics, and more fully resolve phylogenetic relationships among this group, will be invaluable to achieving these diverse goals.

Supplemental Information

Supplemental Information 1 Supplementary tables: Mitogenome annotations.

Supplemental Information 2 Supplemental scripts for de novo genome assembly and IQ-tree.

Supplemental Information 3 Supplementary Figures.

Maximum likelihood phylogeny of 19 antipatharian taxa, and maps of the complete mitochondrial genomes.

We extend heartfelt appreciation to the following individuals who made invaluable contributions to our research: Y. Papastamatiou for samples, C. Lockerman and M. Mizobe at the Hawai‘i Institute of Marine Biology (HIMB) for assistance with library preparation, and J. Saito at the Advanced Studies in Genomics, Proteomics, and Bioinformatics at the University of Hawai‘i for sequencing services. We also thank three anonymous reviewers for comments that helped improve this manuscript. This is contribution #1990 from the Hawai’i Institute of Marine Biology and #11944 from the School of Ocean and Earth Science and Technology at the University of Hawai’i at Mānoa.

Additional Information and Declarations

Competing Interests

Robert J. Toonen is an Academic Editor for PeerJ.

Author Contributions

Van Wishingrad conceived and designed the experiments, analyzed the data, prepared figures and/or tables, authored or reviewed drafts of the article, and approved the final draft.

Leah E. K. Shizuru conceived and designed the experiments, performed the experiments, analyzed the data, prepared figures and/or tables, authored or reviewed drafts of the article, and approved the final draft.

Kenji Takata performed the experiments, analyzed the data, authored or reviewed drafts of the article, and approved the final draft.

Anthony D. Montgomery conceived and designed the experiments, authored or reviewed drafts of the article, and approved the final draft.

Daniel Wagner conceived and designed the experiments, authored or reviewed drafts of the article, and approved the final draft.

Robert J. Toonen conceived and designed the experiments, authored or reviewed drafts of the article, and approved the final draft.

Field Study Permissions

The following information was supplied relating to field study approvals (i.e., approving body and any reference numbers):

All specimens were gathered in accordance with the relevant collection permits issued by co-management agencies of the NOAA Papahānaumokuākea Marine National Monument, as well as the State of Hawai‘i Department of Land and Natural Resources, Division of Aquatic Resources, Special Activities Permits #SAP-2008-04 and SAP-2009-13.

DNA Deposition

The following information was supplied regarding the deposition of DNA sequences:

The genomic sequences are available at GenBank: PP498838, PP506421, PQ189023, PQ189024, PQ189025, ON653414, PQ189026, OP104910, and PQ189027.

The raw sequence reads are available at NCBI: PRJNA1177630.

Data Availability

The following information was supplied regarding data availability:

The code and scripts are available in the Supplemental Files.

1 Portions of this text were previously published as part of a thesis (Shizuru, 2023).

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
