# Peer review of "Hawaiian black coral (Antipatharia) complete mitochondrial genomes have limited phylogenetic signal for taxonomic resolution of species"

_PeerJ, doi:10.7717/peerj.18731_

## Round 0.1 · original submission · Major Revisions

· Academic Editor

Major Revisions

Please respond to all the reviewers’ comments in a point-by-point manner.

Reviewer 1 ·

Basic reporting

The report made on this ms is impeccable. My only issue is excessive self-citation and misinterpreted in-text citations that I specifically highlight in the additional comments.

Experimental design

I have no major issues on the methodology outlines in this ms. Perhaps my only comment is that the ms worked on the complete mitogenomes of "shallow-water" (to mesophotic) black corals in Hawaii.

Validity of the findings

My only comment is for the authors to avoid statements such as "surprising" and the like for findings that have already been supported by previous studies that utilized PCR-amplified genes, mitogenomes, and ultraconserved elements. For example, the polyphyly of the whip black corals is not surprising. Regardless, the findings presented in this paper is important as it provides a complete survey of mitogenomes from shallow-water black corals (<150m).

Additional comments

Specific Comments:

Abstract:
1. Line 29. Maybe emphasized that the work is on “shallow-water and mesophotic” Hawaiian black corals.
2. Line 34. The divergence of Cirrhipathes cf. anguina is not “surprising” as stated in the abstract. It can be restated that this observation is congruent with other studies that utilized PCR-amplified genes and genomic studies.
3. Line 44. Not sure if it is appropriate that the keyword ‘molecular cock’ be included in this paper as I have not read or seen portions of the ms that deduce the divergence of each Hawaiian black corals in geological time.

Introduction:
1. Line 53. Not sure if the word “surface” would be appropriate. Maybe a more appropriate one would be to mention how shallow can they be seen. And maybe include an in-text citation for that one as well similar to 8,600 m (Molodtsova et al. 2008). Some black corals reported in the shallowest zone are those reported in Trinidad at 5 m (Warner 1981), New Zealand (Grange 1985) and Hongkong (Chui and Ang 2015) at 4 meters, and recently at 3 m in the Galapagos (Agarwal et al. 2024). Surely, they can also be seen as shallow as 2 m (especially if the water is turbid) but I cannot locate a single reference.
2. Lines 61-62. The authors misinterpreted the findings of Bo et al. (2019) where in this submitted ms they mentioned “About 63% of antipatharians species”, I want to provide correction to the authors that what Bo et al. (2019) mentioned was “About 63% of the known antipatharian GENERA”. I suggest the authors checked with Molodtsova et al. (2022) that mentioned at least 31.6% antipatharian species (90 species from 27 genera) are described below 800 m. From here, the authors can estimate how many species are in the shallow-water and/or mesophotic depths.
3. Lines 120. The statement “and the original species descriptions are vague” is not supported in Wagner (2015). The original species description of M. ulex is found in Ellis & Solander (1786) while the original species description of C. anguina is found in Dana (1846). The descriptions found in these old monographs are ‘vague’ based on current standard and modern terminologies.
4. Lines 121. In strict terms, “species descriptions” means a formal scientific description of undescribed species new to science. None of the in-text citations (Wagner et al. 2010; Wagner 2015) are “species descriptions.” I can cite few papers that provided ‘species description’ of Hawaiian black corals these include Opresko et al. (2009) – Antipathes griggi, Opresko & Wagner (2020) – Alternatipathes venusta, Umbellapathes litocrada, Antipathes sylospongia, Wagner & Opresko (2015) – Leiopathes anosa. But the latter species descriptions might be out of scope as the present ms deals with shallow-water Hawaiian black corals not deep-sea. But I saw that the authors cited these literatures.
5. Line 135. Please emphasized that the ms ‘present the most complete survey of “shallow-water” Hawaiian black coral phylogenetics”. It did not deal with other deep-sea Hawaiian black corals like Alternatipathes venusta, Umbellapathes litocrada, Leiopathes anosa, Antipathes slyospongia, etc.

Materials & Methods
1. Lines 145-148. Maybe it would be appropriate to mention that the method of collection is described in Wagner (2015). But if my assumption is wrong that the specimens used are not the same as those with Wagner (2015), then kindly disregard.

Results
1. Line 224 "Hexacorallia" to "hexacorallia anthozoans"
2. Lines 224-226. Maybe it’s best to place the similarities between the present results and that of previous studies in the Discussion.
3. Line 246 Best placed this in the Discussion "as is largely agreed upon (Horowitz et al. 2020; Horowitz et al. 2023)”
4. Lines 247-248. The findings that ND5 as the most variable gene region would greatly benefit by comparing it to the findings of Barret et al. (2020) that determined ND4 as the most variable gene region. The authors might want to also consider Horowitz et al. (2020) that mentioned nad5-IGR-nad1(igrN) produced similar topology to that of UCEs.
5. Line 255. Maybe it’s best to place “previous studies (Barrett et al. 2020; Horowitz et al. 2023)” in the Discussion.
6. Line 261-262. It’s also worth mentioning that apart from recovering Myriopathidae as monophyletic (which is expected based on previous studies), that Clade A also consisted of the genus Tylopathes from the Stylopathidae.

Discussion
1. Line 268 emphasized that the work is "among shallow-water 'ekaha ku moana' (Hawaiian black coral species"
2. Line 270 “Our analysis of all the known diversity of Hawaiian black corals” It should be mentioned that what was analysed were species from the shallow-water to mesophotic zones.
3. Line 276. I think that the reference Wagner (2015) might be the paper published in Front. Mar. Sci., not the paper currently placed in the references which is on the spatial distribution of Hawaiian black corals.
4. Line 280. Wagner (2015) identified the specimen as "Stichopathes sp.?" but due to the morphological description similar to that of S. maldivensis, later scholars referred the Hawaiian specimen as "Stichopathes maldivensis" (i.e. Terrana et al. 2020; Bacharo & Sotto 2022). Do the authors agree with the previous scholars that synonymized the Hawaiian Stichopathes specimen as S. maldivensis?
5. Lines 282-284; 286-287. The description of Family Aphanipathidae is provided in Opresko (2004) alone. Unless Wagner (2015) emend it. I see no reason to cite the latter.
6. Lines 289-293. Likewise, the description of the Family Myriopathidae is provided in Opresko (2001) alone. I’m not sure if Wagner (2015) made any emendment to Myriopathidae. If not, I see no reason for it to be cited for the description of Myriopathidae.
7. Line 294-298. Those in the know would probably understand why there is no sufficient information available to discern the taxonomic characters of the Family Antipathidae which can be attributed to its morphologically variable characters (with almost no diagnostic character). For example in Daly et al. (2007), when the seven black coral families were introduced, only Antipathidae was not clearly distinguished as to what morphologies is unique to them. This is similar to Opresko (2005) (New species from Southern California Bight), where the morpho-taxonomy of Antipathidae was not clearly described. I apologize if I am not in consolidation with the description of Antipathidae placed specific lines, specifically describing the antipathidae polyps as "0.5 to 1.0 mm in transverse diameter". Whip black corals [technically still] belonging to the family Antipathidae has among the largest polyps in the entire order where documented findings reported that it can reach 3.9 mm in size (Terrana et al. 2020).
8. Line 301-302. I agree with these specific lines. But other antipathid genera that can be distinguished in-situ would include Allopathes, that has multiple elongated stems arising from the base (Opresko 1994), Blastopathes, and its stem-like branches (Horowitz et al. 2020). The rest which include Aracnopathes is almost similar to Antipathes, and some of the genera like Pteropathes and Hillopathes are based on line drawings which are 'vague'.
9. Lines 305-310. None of the multiple references provided included the taxonomic changes made to the genus Antipathes and the Family Antipathidae. I can provide some relevant in-text citations for this claim which would include the previously identified Antipathes that has systematically arranged ramifications of equal length (=pinnules) which was later transferred to Myriopathes (Opresko 2001). Another relevant reference for the claim include the deep-water (>1000 m) Stichopathes that has now been re-classified as Aphanostichopathes under Family Aphanipathidae (Opresko et al. 2021).
10. Provide example of genus from Antipathidae that have been removed. Such as for example Stichopathes deeper than >1000 m are now re-classified to Aphanostichopathes (see Opresko et al. 2021), and maybe Myriopathes-like specimens desctibed as Antipathes americana (from Opresko 1972) that have been elevated to a new family Ameriopathidae (Horowitz et al. 2024). Another classic example is those worked by Opresko (2001) that separated all previously identified Antipathes and placed it under Family Myriopathidae.
11. Line 309. Perhaps the elephant in the room in Antipatharian taxonomy is the type species Antipathes dichotoma that has been showed to fall under Aphanipathidae than Antipathidae.
12. Line 313. Wagner (2015) that worked with spatial distribution of Hawaiian black corals (as currently referenced to) has no molecular data (even the Wagner 2015 Front. Mar. Sci.) to support the statement "Previous authors have found molecular data do not support existing morphologically-based taxonomy"
13. Lines 315-316. I am confused what the authors try to convey regarding the lines “Here, we confirm that Myriopathes cf. ulex is closely related to Myriopathes ulex”, as it is something surprising.
14. Line 319. Wagner (2015) was not only one who used the statement “taxonomic dumping ground” to describe Antipathidae. It was also used in the papers of Brugler et al. (2013), Bo et al. (2018), etc. Perhaps the statement can be traced as far as in Daly et al. (2007).
15. Line 328. Duminil & Besnard 2021 is not provided in the reference.
16. Lines 329-330. I understand that the in-text citation for Antipatharia and Zoantharia are those by Barret et al. (2020) and Sinniger et al. (2007), respectively. How about the in-text citations for Actinaria and Scleractinia?


References
1. I think the reference for Wagner (2015) is missing:
Wagner, D. (2015). A taxonomic survey of the shallow-water (< 150 m) black corals (Cnidaria: Antipatharia) of the Hawaiian Islands. Frontiers in Marine Science, 2, 24.
2. Duminil & Besnard 2021 is missing in the References.
3. Line 558. Opresko (2004) has an incomplete bibliographic data. The title should include “Establishment of a new family, Aphanipathidae”

Reviewer 2 ·

Basic reporting

- The word “presumed” was used twice on line 103. Perhaps you can find an alternative word for second use of the term.
- The phrase “to date” was used in subsequent lines, 133 and 135. It is suggested to find an alternative phrase for the second usage.
- There is a long section discussing the cultural relevance of black corals (lines 66-73) and black coral fishery in Hawaii (lines 80-114). How does this address a specific objective or hypothesis? Is this important in providing an appropriate background for your reported genomic results? It is recommended to instead provide a more focused introduction underlining the relevance of genomes that anticipates the insights you discussed related to the systematics and taxonomy of Hawaiian black corals.

Experimental design

No Comment

Validity of the findings

- The highlighted results explained in the abstract are not explicitly reflected in the conclusion.
- Would it be possible in your paper to articulate a more robust set of conclusions that characterize and focus on the general aspects of your findings on the Hawaiian black corals’ complete mitochondrial genomes.
- How does the statement on lines 375-377, “The recent discovery that there is no depth refuge for the most targeted species, as once believed, raised significant concerns about the long-term sustainability of the moana fishery “(which has been cited from previous work) connects with the reported data in your manuscript?

Reviewer 3 ·

Basic reporting

I only have a few questions and suggestions regarding the main and supplemental figures - likely with only minor revisions. And additionally some clarifications for the introduction. These are further addressed in the attached pdf narrative.

Experimental design

Minor revision in the methods regarding collection location and time. This is further addressed in the attached pdf narrative.

Validity of the findings

no comment

Additional comments

Please see the attached pdf for summary, specific line number question/comments, and selected suggestions.

Annotated reviews are not available for download in order to protect the identity of reviewers who chose to remain anonymous.

---

## Round 0.2 · Minor Revisions

· Academic Editor

Minor Revisions

Please revise the manuscript by following the reviewer's comments.

Reviewer 1 ·

Basic reporting

As I mentioned prior, the ms is impeccable in its initial stage and has been refined better in the current version. To add, I want to mention that I fully support the comment made by Reviewer 3 that while the Introduction contained cultural importance of Hawaiian black corals which has no 'direct' connection to the subject matter (=mitogenomes), I find high relevance in mentioning this, and I think should be the way forward for our field. Many recent papers on black corals from the Red Sea, South China Sea, Madagascar, India, the Philippines, Australia, etc. have been published but have not included a cultural perspective of specimens collected. I am glad that the authors retained that particular part of the Introduction in the second version.

Experimental design

One of my major comments was to reframe the scope to "shallow-water to mesophotic" were agreed and adhered to by the authors.

Validity of the findings

The "softening of the findings" were greatly appreciated, and it is important to note while this study (that utilized mitogenomes) has limitations as implied in the title, the findings has important implications to the advancement of black coral systematics and evolution.

Additional comments

1. I just want to mention that the PDF and Track Changes version are not similar. With the former being more updated.
2. Line 50. Do we really need to 'newly' insert the in-text citation by Shizuru (2023) for the claim? This claim is popular across any antipatharian literature and has been provided/reviewed in greater details in the papers of Daly et al. (2007), Bo et al. (2008), Wagner et al. (2012), and Brugler et al. (2013). Which I want to point that these literatures have been used throughout the ms.
3. Lines 61-63. I would suggest to omit the word ‘primary’. While it is true that black corals provide structural framework at these depths but there are also octocorals and sponges that provide the same (or arguably even more) three-dimensionality to the benthos of the mesophotic zone.
4. Line 121-122. Rephrase “For example, type materials are missing for Cirrhipathes anguina and M. ulex (Wagner 2015), and the original species descriptions of these species are vague (Ellis & Solander 1786; Dana 1846).” Notice that I removed Wagner (2015) from the in-text citation of original species descriptions.
5. Line 124-125. I am glad that while the authors made some changes to Line 122. The in-text citation in this particular statement should be changed. Again, I want to reiterate Wagner (2010, 2015) did not contain any SPECIES DESCIRPTIONS (=strictly means undocumented species NEW to science). Hence, it should be: “Recent species descriptions have relied predominantly on taxonomic characters such as colony branching pattern, polyp structure, and skeletal spine morphology, as well as in situ photographs and scanning electron microscopy of skeletal features (Opresko et al. 2009; Wagner & Opresko 2015; Opresko & Wagner 2020).”
6. Line 123-125. If the authors are persistent in keeping the original statement which includes the word “species description” which I want to clarify (for the second time) to the authors that it is not TAXONOMICALLY sound, then they can rephrase it by mentioning “Recent species documentation of black corals in Hawaii have relied …” (notice that I change species description to ‘species documentation’)
7. Line 277. Since it has been newly added in the Reference section. You can add Horowitz et al. (2020) in the in-text citation for UCE studies as well.
8. Line 285. This line is different from what was currently written in the PDF and the track changes version. I find the statement written in the track changes version better: “later synonymized as Stichopathes maldivensis (REF)” than what is written in the PDF file.
9. The authors may also want to utilize the “Comments” section in Table 1 where they could place the synonymy of the Hawaiian Stichopathes? to S. cf. maldivensis by other studies. Preferably, I would mention that the initial identification was “Stichopathes? sp” with the question mark (similar to what was written in Wagner (2015)) which was later synonymized by succeeding authors similar to the Stichopathes Clade D.
10. Line 299-306. This is just a punctuation comment. There are a lot of parentheses in these lines. I think Lines 302-303 was not sealed/closed with a “right parenthesis )“
11. Line 312. Maybe better to restate as “but also note Allopathes and Blastopathes have not been reported in Hawai’I”. This is to indicate that MAYBE these species are thriving elsewhere (apart from their type locality), and maybe they occur in Hawai’a also? But not encountered yet.
12. Line 315. ‘Historically’ a taxonomic dumping ground. With the current integrated taxonomic approach, ‘dumping’ undescribed species to Antipathes is no longer rampant as it was. I want to point our that the description "dumping ground" refers to a particular clade/taxa (in this case Antipathidae) that has been dumped with species that cannot be associated with other clade/taxa. Kindly manage this similar writing tone in other parts of the ms.
13. Line 318. Changed “equal length” to “systematically arranged”
14. Line 323-324. I think this was a new addition to the PDF that was not found in the track changes version? Anyway, I agree with this statement but kindly rephrase this sentence since there are two “Reliable” words that are found in the start and end of the sentences. Maybe delete the “Reliable” at the beginning of the sentence.
15. Line 329. If the authors agree, change “molecular data” to “mitogenomic data”
16. Line 358. I understand that the authors are in the same research group as the previous published papers in Hawaii. But would the authors agree that the statement “including the first representatives from the genera … Cirrhipathes.” is TECHNICALLY NOT true? My point is that the first mitogenome representative from a Cirrhipathes species was published by Shizuru et al. (2023), not by Wishingrad et al. (under review). So only Aphanipathes is technically a NEW representative in this paper. I hope I made a point.
17. Line 360-361. I find it weird to mention “it is not possible to say whether mitogenome size may be a diagnostic character, but this remains an intriguing possibility”, I suggest that this be DELETED as it contradicts Lines 385-386: "family level and limited resolution at the species level for ekaha ku moana based on these mitogenomes, it will likely require genomic-wide sequencing to resolve these relationships."
18. Figure 1. The caption mentioned about “Table S1” which was missing or not attached as part of the Supplementary materials. I want to check if everything has been updated there too.
19. Figure 1. The caption is still “Stichopathes sp”
20. Figure 2. The “cf” should NOT be italicized.
21. Figure 2. Shouldn’t the description for specimen #269 should be “Myriopathes cf. ulex” with the “cf”
22. Figures S1, S2, S3, S4, S5, S6, S7. The description for Stichopathes cf. maldivensis is still Stichopathes sp.
23. Figure S16. The description is still Stichopathes sp.
24. The identity of the Stichopathes sp found in the supplementary file containing the mitogenome annotations should also be changed.


Kindly incorporate some of these minor comments and I have congratulate the authors in advance for this impressive published work!

Reviewer 3 ·

Basic reporting

No comment

Experimental design

No comment

Validity of the findings

No comment

Additional comments

I thank the authors for addressing my previous edits / suggestions and have nothing further to add.

---

## Round 0.3 · accepted · Accept

· Academic Editor

Accept

The authors responded well to the reviewer's comments, and the manuscript can be accepted for publication now.